# Successive redox-mediated visible-light ferrophotovoltaics

Yuji Noguchi [1✉], Yuki Taniguchi[1], Ryotaro Inoue[2] & Masaru Miyayama [1]

Titanium oxide materials have multiple functions such as photocatalytic and photovoltaic effects. Ferroelectrics provide access to light energy conversion that delivers above-bandgap voltages arising from spatial inversion symmetry breaking, whereas their wide bandgap leads to poor absorption of visible light. Bandgap narrowing offers a potential solution, but this material modification suppresses spontaneous polarization and, hence, sacrifices photovoltages. Here, we report successive-redox mediated ferrophotovoltaics that exhibit a robust visible-light response. Our single-crystal experiments and ab initio calculations, along with photo-luminescence analysis, demonstrate that divalent $Fe^{2+}$ and trivalent $Fe^{3+}$ coexisted in a prototypical ferroelectric barium titanate $BaTiO_3$ introduce donor and acceptor levels, respectively, and that two sequential $Fe^{3+}/Fe^{2+}$ redox reactions enhance the photogenerated power not only under visible light but also at photon energies greater than the bandgap. Our approach opens a promising route to the visible-light activation of photovoltaics and, potentially, of photocatalysts.

[1] Department of Applied Chemistry, School of Engineering, The University of Tokyo, Bunkyo-Ku, Tokyo 113-8654, Japan. [2] Division of Physics, Institute of Liberal Education, School of Medicine, Nihon University, Tokyo 173-8610, Japan. ✉email: yuji19700126@gmail.com

Light-induced functions such as photocatalytic and photo-voltaic (PV) effects rely on the formation of electron ($e'$)-hole ($h^\bullet$) pairs under illumination. As a representative photocatalyst, titanium dioxide[1], absorbs only ultraviolet light owing to its large bandgap ($E_g$), several approaches for controlling the electronic structure have been used to activate a visible-light response, often with the aid of sacrificial reagents, dyes or precious metal co-catalysts: introducing donor levels into the bandgap via doping[2,3], raising the valence band to a more negative level than that arising from O-$2p$ by a composition modification[4,5], and narrowing the bandgap by forming a solid solution[6] or a two-dimensional structure[7].

For strontium titanate, Reunchan et al.[8] have demonstrated that the Fermi-level lifting by a co-doping promotes a photo-catalytic response under visible-light, which is achieved by sta-bilizing $Cr^{3+}$ in the presence of $La^{3+}$. Defect levels inside the bandgap, called gap states, of $Cr^{3+}$ are located above the valence band maximum (VBM), and the donor states play a crucial role in visible-light activation[9]. Wang et al.[10] have shown that La- and Rh-codoped $SrTiO_3$ functions as a photocatalyst, where the occupied states of $Rh^{3+}$ narrows the bandgap.

Ferrophotovoltaics have attracted considerable interest because their bulk PV effect[11–13] leads to high voltages that are beyond the bandgap limit of semiconductor p–n junctions[11,12,14,15]. The fer-roelectric PV effect has been extensively studied for wide-bandgap oxides such as $LiNbO_3$[11,16,17], $BaTiO_3$[18–22], $PbTiO_3$-based per-ovskites[23–25], and $BiFeO_3$[14,15,26]. The poor absorption of visible-light is partially overcome by narrowing the bandgap[27–31]. This is realized by incorporating non-polar components into the widegap ferroelectrics, which is, in principle, accompanied by a substantial decrease in spontaneous polarization ($P_s$). As the intense photo-response originates from the large magnitude of $P_s$[13,32], these material modifications are inevitably associated with a marked suppression of photovoltage. Gap-state engineering that utilizes mid-gap states for $BiFeO_3$[33] can enhance photocurrents without sacrificing photovoltages, where the photon energy of at least half of $E_g$ are required.

Here, we report successive-redox-mediated ferrophotovoltaics, where two gap states realize $e'$-$h^\bullet$ pair formation over a wide photon-energy ($h\nu$) range. Our approach is based on acceptor and donor states that act as scaffolds for generating photoinduced carriers. A single transition-metal dopant with two different valence states introduces these gap states, thereby providing successive redox cycles under illumination. In principle, it can induce PV currents at small $h\nu$ without being restricted by the material bandgap. Our experimental and theoretical study on a prototypical ferroelectric $BaTiO_3$ demonstrates that the $3d$ orbi-tals of iron derive donor and acceptor levels in the $Fe^{2+}$-$Fe^{3+}$ coexisting state and that $e'$-$h^\bullet$ pairs injected by two sequential $Fe^{3+}$/$Fe^{2+}$ reactions deliver a robust PV response not only under visible-light but also at $h\nu$ greater than $E_g$.

## Results

**Strategy for visible-light activation.** Because of the wide $E_g$ of $BaTiO_3$ (3.2 to 3.3 eV)[34,35], the PV effect occurs under ultraviolet light. To activate a visible-light response, we introduce electronic states derived from $Fe^{2+}$ and $Fe^{3+}$ into the bandgap and utilize these gap states as scaffolds for carrier generation. The most striking feature is that $e'$-$h^\bullet$ pairs are, in principle, formed at photon energies much smaller than that of the mid-gap-state engineering[33], as described later.

In $O_h$ symmetry (Fig. 1a, b), the electronic configurations of iron in the high-spin state are expressed as $Fe^{3+}$ ($d^5$) with $t_{2g}^3$(up) $e_g^2$(up) $t_{2g}^0$(down) $e_g^0$(down) and $Fe^{2+}$ ($d^6$) with $t_{2g}^3$(up) $e_g^2$(up) $t_{2g}^1$(down) $e_g^0$(down)[36]. In the $BaTiO_3$ lattice, the states of $t_{2g}^3$

(up) $e_g^2$(up) are located near the bottom of the valence band[37], while it is probable that the states of $t_{2g}^0$(down) of $Fe^{3+}$ (Fig. 1c) and $t_{2g}^1$(down) of $Fe^{2+}$ (Fig. 1d) are present inside the bandgap by tuning their local structures. We, therefore, consider the following strategy for generating $e'$-$h^\bullet$ pairs under visible-light: $Fe^{3+}$ plays the role of an electron acceptor that results in hole injection into the valence band[38,39], and $Fe^{2+}$ acts as an electron donor that leads to electron injection into the conduction band[40–42]. Provided that $Fe^{3+}$ and $Fe^{2+}$ coexist and also that their acceptor and donor states are present at moderate levels near the VBM (the top of the O-$2p$ band) and the conduction band minimum (CBM: the bottom of the Ti-$3d$ band), respectively, $e'$-$h^\bullet$ pairs can be created at $h\nu$ much smaller than $E_g$. As displayed in Supplementary Fig. 1, the hybridization between Fe-$3d$ and its adjacent orbitals provides the bonding states ($t_{2g}$ and $e_g$) and the antibonding states ($t_{2g}^*$ and $e_g^*$), as for the majority spin components. Hereafter, the Fe-$3d$ derived states without an asterisk, such as $t_{2g}$ and $d_{xy}$, are the bonding states, while those with asterisk ($*$), e.g., $t_{2g}^*$ and $d_{xy}^*$, are the antibonding ones.

Figure 1e plots the defect concentrations at 25 °C as a function of $Po_2^{900\,°C}$ in Fe (0.3%)-doped $BaTiO_3$. These calculations are based on the assumption that the oxygen vacancy concentration, $[V_O^{\bullet\bullet}]$, equilibrated in a high-temperature state at 900 °C remains unchanged at 25 °C ($Po_2^{900\,°C}$ is the oxygen partial pressure at 900 °C). The defect feature is divided into four regions. In the regions I and II, the majority of iron is trivalent, namely, the concentration of $Fe^{3+}$, $[Fe^{3+}]$, is several orders of magnitude higher than others; the minority is $Fe^{4+}$ at a higher $Po_2$ (I) and $Fe^{2+}$ at a lower $Po_2$ (II). We note that the region III is regarded as the coexisting state of $Fe^{3+}$ and $Fe^{2+}$, the concentrations of which have the same order. At the higher $Po_2^{900\,°C}$ side (I, II, and III), the charge neutrality is approximated as $2[Fe^{2+}] + [Fe^{3+}] \approx 2 [V_O^{\bullet\bullet}]$. The region IV is semiconducting with a high-electron concentration ($n$), along with $Fe^{2+}$ as the majority, where the charge neutrality is expressed as $n \approx 2[V_O^{\bullet\bullet}]$ except near the transition $Po_2^{900\,°C}$. According to the strategy described above, we expect that the coexisting state of $Fe^{3+}$ and $Fe^{2+}$ in the region III is capable of inducing a robust PV response under visible-light. We therefore adopt an $Po_2^{900\,°C}$ of $10^{-23}$ atm as an annealing condition, where the concentrations are calculated to be $[Fe^{3+}] \approx 3.5 \times 10^{19}$ cm$^{-3}$ and $[Fe^{2+}] \approx 4.3 \times 10^{19}$ cm$^{-3}$.

**Formation of defect associates.** Figure 2 displays the DFT energy versus $n$ of $V_{On}^{\bullet\bullet}$, where $V_{On}^{\bullet\bullet}$ denotes the oxygen vacancy on the $n$-th nearest-neighbor site with respect to Fe (Supplementary Fig. 2). The cells with $n = 1$ to 3 exhibit small energies compared with those with $n > 4$ regardless of the Fe valence. Considering the energy difference of 0.2 to 0.8 eV, we posit that an attractive interaction is formed between iron and $V_O^{\bullet\bullet}$. If $V_O^{\bullet\bullet}$ is mobile below the Curie temperature ($T_C$), we consider that $V_O^{\bullet\bullet}$ is trapped by Fe and is eventually positioned on the nearest-neighbor O1 site (Fig. 3b, c) after a certain period of time. These results suggest that $V_O^{\bullet\bullet}$ exists as a defect associate of $Fe^{3+}$-$V_O^{\bullet\bullet}$ and/or $Fe^{2+}$-$V_O^{\bullet\bullet}$.

The formation of the defect associates can be explained in terms of the energy levels of the Fe-derived states. The orbital interactions provide bonding and antibonding states in both the majority ($\uparrow$) and minority ($\downarrow$) spin bands. In the $\uparrow$ band, the bonding states appear near the bottom of the valence band (Supplementary Fig. 3), and the antibonding states (marked with asterisk) arise near or inside the bandgap (Fig. 4 and Supplementary Fig. 4). For the $Fe^{3+}$-$V_{O1}^{\bullet\bullet}$ and $Fe^{3+}$-$V_{O4}^{\bullet\bullet}$ cells, the $t_{2g}^*$ and $e_g^*$ states are similar and their energy levels are close, while the $t_{2g}$ and $e_g$ states exhibit different features (Supplemen-tary Fig. 3): these bonding states of the $Fe^{3+}$-$V_{O1}^{\bullet\bullet}$ cell are lower

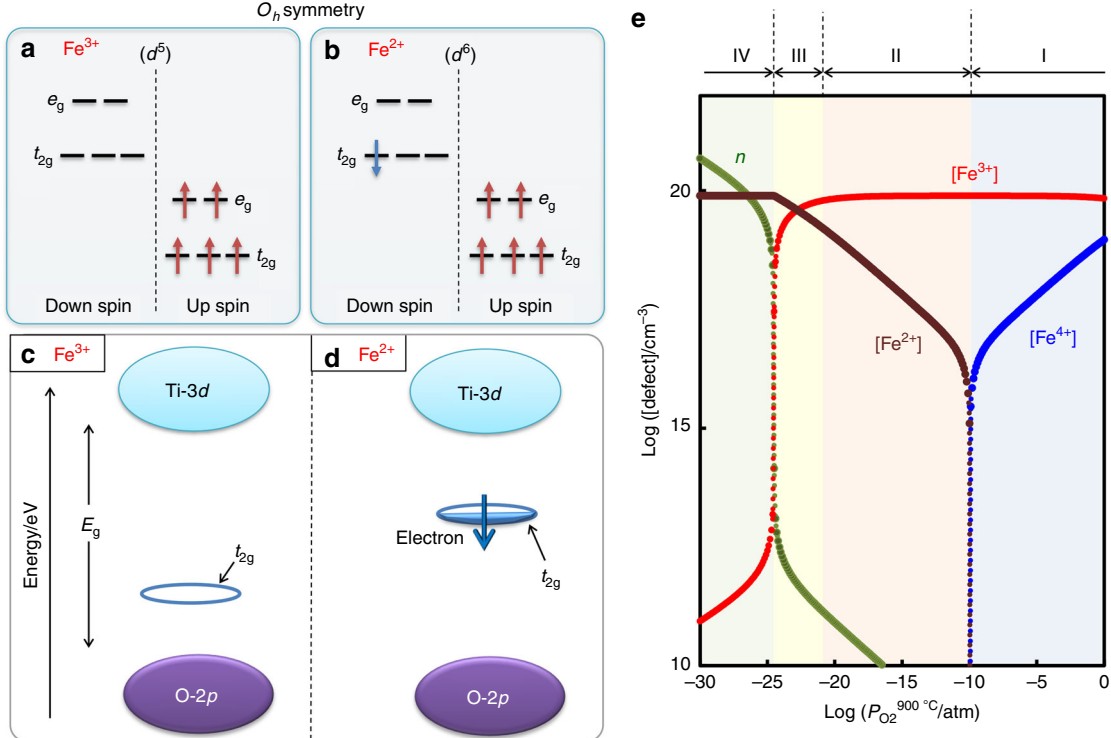

**Fig. 1 Defects states in Fe-doped BaTiO₃. a, b** Electronic configurations in the high-spin state of $Fe^{3+}$ and $Fe^{2+}$ in $O_h$ symmetry. **c, d** Gap states derived from $Fe^{2+}$ and $Fe^{3+}$ in the bandgap ($E_g$) between the conduction band (Ti-3d) and the valence band (O-2p) in the host BaTiO₃ lattice. **e** Defect concentrations at 25 °C as a function of oxygen partial pressure ($P_{O_2}^{900 °C}$) at 900 °C (annealing temperature). In the region III, $Fe^{2+}$ and $Fe^{3+}$ coexist and their defect (gap) states can be utilized as scaffolds for visible-light absorption. In the $Fe^{2+}$-$Fe^{3+}$ coexisting state, photoinduced carriers are expected to be generated as follows: the unoccupied $t_{2g}$ state of $Fe^{3+}$ **c** is positioned above the valence band maximum (VBM, the top of the O-2p band) and then play the role of an electron acceptor, while the electron-occupying $t_{2g}$ state of $Fe^{2+}$ **d** is located below the conduction band minimum (CBM, the bottom of the Ti-3d band) and then acts as an electron donor.

in energy by ≈0.5 eV than those of the $Fe^{3+}$-$V_{O4}^{••}$ cell. The attractive interaction between $Fe^{3+}$ and $V_O^{••}$ originates from these low-lying bonding states. As for $Fe^{2+}$, the $Fe^{2+}$-$V_{O4}^{••}$ cell (Supplementary Fig. 4c, d) has electron-occupying gap states: the Fe-$3d_{xz}$ state (↓) and the Fe-$3d_{z^2}^*$, -$3d_{x^2-y^2}^*$ and -$3d_{yz}^*$ states in the ↑ band. In contrast, the $Fe^{2+}$-$V_{O1}^{••}$ cell (Fig. 4e, f,) exhibits only two gap states of Fe-$3d_{z^2}$ (↓) and -$3d_{x^2-y^2}^*$ (↑). The remaining ↑ states, namely, Fe-$3d_{xz}$, -$3d_{yz}^*$, -$3d_{xy}^*$, and -$3d_{z^2}$, are lower in energy and are near the bottom of the valence band. The stabilization of $V_O^{••}$ on the O1 site, i.e., the formation of the $Fe^2$-$V_O^{••}$ associate, results from these lower-lying Fe-3d states.

As a short-range $V_O^{••}$ motion with a distance of a few unit cells occurs even near room temperature[43,44], it is reasonable to assume that all oxygen vacancies are present as the defect associates of $Fe^{3+}$-$V_O^{••}$ and/or $Fe^{2+}$-$V_O^{••}$ in our samples, as reported for SrTiO₃[45,46] and PbTiO₃[47,48] and BaTiO₃[49,50]. Considering the charge neutrality and the strong attractive interaction between iron and $V_O^{••}$ (Fig. 2), we can say that the $Fe^{2+}$-$V_O^{••}$ associate[51] is the majority, i.e., the concentration of [$Fe^{2+}$-$V_O^{••}$] is several orders or magnitude higher than that of an isolated $Fe^{2+}$ (Fig. 3c). Similarly, the $Fe^{3+}$-$V_O^{••}$ associate[51,52] (Fig. 3b) and an isolated $Fe^{3+}$[53] are present with approximately the same concentration (Fig. 3a).

**Gap states derived from Fe-3d.** Figure 4 displays the electronic structures of the $Fe^{3+}$, $Fe^{3+}$-$V_O^{••}$, and $Fe^{2+}$-$V_O^{••}$ cells, the crystal structures of which are shown in Fig. 3. In Fig. 5, the defect levels for the $Fe^{2+}$-$V_{O1}^{••}$ and $Fe^{3+}$-$V_{O1}^{••}$ cells are illustrated along with

the corresponding partial charges of the Fe-3d derived gaps states. As reported in the literature[54,55], the valence band is mainly formed by O-2p, while the conduction band is primarily composed of Ti-3d, where an orbital hybridization between them occurs.

Here, we focus on the electronic configuration and the defect levels derived from Fe-3d. In the $Fe^{3+}$ cell (Fig. 4a, b), the empty $t_{2g}$ (↓) state appears in the bandgap at a depth of 1.9–2.3 eV from the VBM. For the $Fe^{3+}$-$V_O^{••}$ cell (Fig. 4c, d), four empty bonding (↓) states arise near the middle of the bandgap; at the Γ point, two are low-lying degenerate states of Fe-$3d_{xz}$ and -$3d_{yz}$ (Fig. 5e), and the remaining two are higher-lying Fe-$3d_{z^2}$ (Fig. 5f) and -$3d_{xy}$ (Fig. 5g) states. The energy difference between these gap states and the VBM is 1.8–2.1 eV. Both for the single isolated $Fe^{3+}$ and the $Fe^{3+}$-$V_O^{••}$ associate (Fig. 5d), we consider that visible-light at $h\nu$ above 1.8 to 1.9 eV can pump electrons from the VBM to the empty states, namely, visible-light drives $h^•$ injection from $Fe^{3+}$ to the valence band. This is achieved by an electron transfer to $Fe^{3+}$ from the adjacent atoms, which is accompanied by a reduction from $Fe^{3+}$ to $Fe^{2+}$.

Figure 4e, f displays the electronic structure of the $Fe^{2+}$-$V_{O1}^{••}$ cell, and a schematic representation is shown in Fig. 5a. There are several Fe-3d derived states in the bandgap: the low-lying two states are electron-occupied, while three states near the CBM are empty. The filled Fe-$3d_{z^2}$ (↓) state (Fig. 5b) emerges in the middle of the bandgap, which is positioned at a depth of 1.5–1.9 eV from the CBM. This energy difference in the vicinity of the k-points of Z–A–R–Z remains unchanged at ≈1.5 eV. Therefore, this occupied state can deliver an onset of electron injection to the

conduction band at this photon energy. As shown in Fig. 5c, the occupied Fe-$3d_{x^2-y^2}$* ($\downarrow$) state with significant dispersion is present. As this state is located at a depth of 2.3–2.6 eV from the CBM, an additional onset of electron injection appears at ≈2.3 eV. These electron pumping processes can be regarded as an electron transfer from $Fe^{2+}$ to titanium along with an oxidation from $Fe^{2+}$ to $Fe^{3+}$.

In the $Fe^{2+}$ and $Fe^{3+}$ coexisting state under illumination, the first onset is at ≈1.5 eV, above which the electron in the $Fe^{2+}$-$3d_{z^2}$ ($\downarrow$) state is pumped to the conduction band (Fig. 5a). This serves as a PV response originating from electron conduction through a

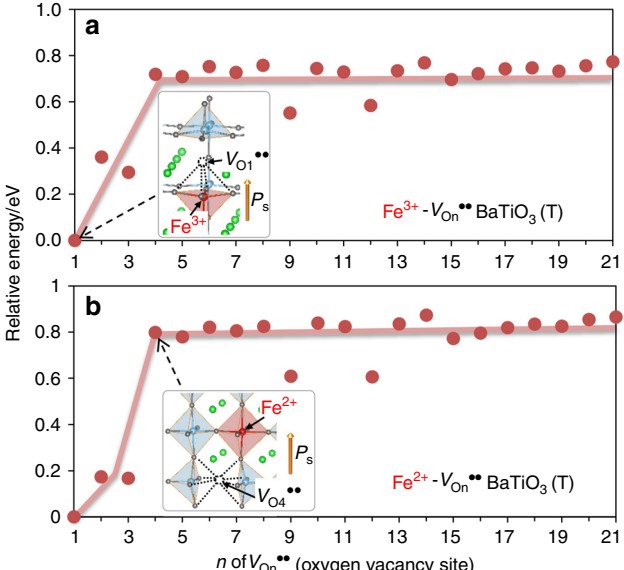

**Fig. 2 DFT energies of Fe-doped BaTiO$_3$.** The horizontal axis $n$ denotes the $V_{On}^{\bullet\bullet}$ (oxygen vacancy)-site number in the tetragonal Ba$_{27}$Ti$_{26}$FeO$_{81}$ structures (Supplementary Fig. 2), and the vertical axis is the density-functional-theory (DFT) energy with respect to that of $n = 1$ in the **a** $Fe^{3+}$ and **b** $Fe^{2+}$ cells. $V_O^{\bullet\bullet}$ is adjacent to iron atoms in the $n = 1$ to 3 cells and is away from iron atoms in the $n = 4$ to 21 cells, as illustrated in the insets of the $n = 1$ ($V_{O1}^{\bullet\bullet}$) and $n = 4$ ($V_{O4}^{\bullet\bullet}$) cells. Regardless of the valence state of iron, the energies with $n = 1$ to 3 are 0.6 to 0.8 eV lower than those with $n = 4$ to 21. These results show that $V_O^{\bullet\bullet}$ is attracted by iron and eventually stabilized on the O1 site (the nearest-neighbor site with Fe).

trapping-detrapping process[16,17,56]. The second onset is at 1.8–1.9 eV, above which holes are injected into the valence band from the empty gap states ($\downarrow$) of $Fe^{3+}$ (Fig. 5d). At $h\nu$ greater than the second onset energy, the photoinduced carrier is regarded as $e'$-$h^\bullet$ pair. The third onset is at 2.3 eV, above which the electron in the $Fe^{2+}$-$3d_{x^2-y^2}$* ($\uparrow$) state is additionally injected into the conduction band (Fig. 5a). We note that $e'$-$h^\bullet$ pairs are effectively generated above the third onset, because visible-light can pump electrons from the valence band to the conduction band mediated through these occupied and empty gap states, namely, electrons are excited from the two occupied states of $Fe^{2+}$ (resulting in an oxidation to $Fe^{3+}$), and holes are injected into the valence band from the empty gap states of $Fe^3$ (leading to a reduction to $Fe^{2+}$). This sequential electron pumping is accompanied by two successive redox reactions of $Fe^{3+}$/$Fe^{2+}$. The PV response mediated through the occupied (donor) and empty (acceptor) states is termed the successive redox-mediated ferrophotovoltaics.

**Visible-light PV properties.** Figure 6 shows the current density normalized by optical intensity, $J_{sc} I_{opt}^{-1}$, as a function of $h\nu$. Both the samples exhibit a complicated PV response above $E_g$ of 3.2 eV. While this behavior is assumed to be explained by the shift current theory[13], we focus on the PV properties at $h\nu$ below $E_g$.

The oxidized (Fig. 6a, b) and non-doped (Supplementary Fig. 5) samples exhibit a PV onset at 1.9–2.0 eV, which is in good agreement with the energy difference between the VBM and the empty states of the isolated $Fe^{3+}$ and the $Fe^{3+}$-$V_O^{\bullet\bullet}$ associate (Fig. 4). As displayed in Fig. 6c, d, the reduced sample indicates a much higher $J_{sc} I_{opt}^{-1}$ in a wide $h\nu$ range. Considering the strong emission lines of the light source (Xe lamp) in the near-infrared region of >820 nm (>1.5 eV, Supplementary Fig. 6) and the low-$h\nu$ data (the inset of Fig. 6d), we posit that the first onset is below 1.6 eV. This agrees with the 1.5 eV difference of the CBM to $Fe^{2+}$-$3d_{z^2}$ ($\downarrow$) (Fig. 5a). The second onset was found at ≈1.9 eV, which is in quantitative agreement with the difference between the VBM and the empty $Fe^{3+}$ ($\downarrow$) states (Fig. 5b). The third onset was observed at ≈2.4 eV, which accords with the gap of the CBM to the filled $Fe^{2+}$-$3d_{x^2-y^2}$* ($\uparrow$) state.

Here, we address the photogenerated carriers under visible-light. The carrier for the oxidized sample is $h^\bullet$ above the first onset, while that for the reduced sample is $e'$ in the $h\nu$ range from the first to the second onset. These single-carrier-type PV effects lead to a $|J_{sc} I_{opt}^{-1}|$ of ≈$1 \times 10^{-9}$ V$^{-1}$ at most. For the reduced sample

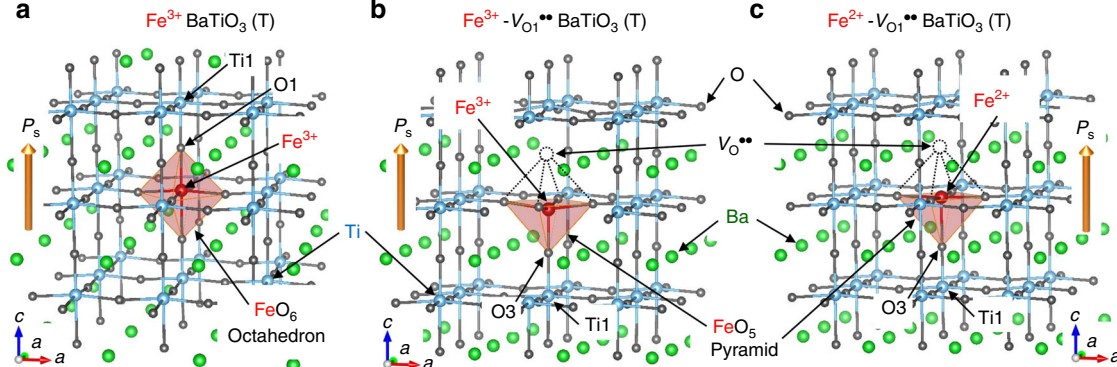

**Fig. 3 Crystal structures of tetragonal (T) Fe-BaTiO$_3$ cells.** The crystal structures optimized by density-functional theory (DFT) calculations are displayed for **a** Ba$_{27}$Ti$_{26}$FeO$_{81}$ ($Fe^{3+}$), **b** Ba$_{27}$Ti$_{26}$FeO$_{80}$ ($Fe^{3+}$-$V_{O1}^{\bullet\bullet}$), and **c** Ba$_{27}$Ti$_{26}$FeO$_{80}$ ($Fe^{2+}$-$V_{O1}^{\bullet\bullet}$). The valence states of Fe are controlled by the total number of electrons. The crystal symmetry was converged in tetragonal space group $P4mm$ for all the cells, where the spontaneous polarization ($P_s$) is parallel to the $c$ axis. O1 and O3 are defined as the first and third nearest-neighbor (NN) oxygen atoms with respect to Fe in the defect-free cell. Ti1 is the first-NN titanium atom with Fe. In the defective cells, an oxygen vacancy ($V_O^{\bullet\bullet}$) is stabilized on the first NN O1 site (see Fig. 2), namely, is present as the associate of $Fe^{3+}$-$V_O^{\bullet\bullet}$ or $Fe^{2+}$-$V_O^{\bullet\bullet}$.

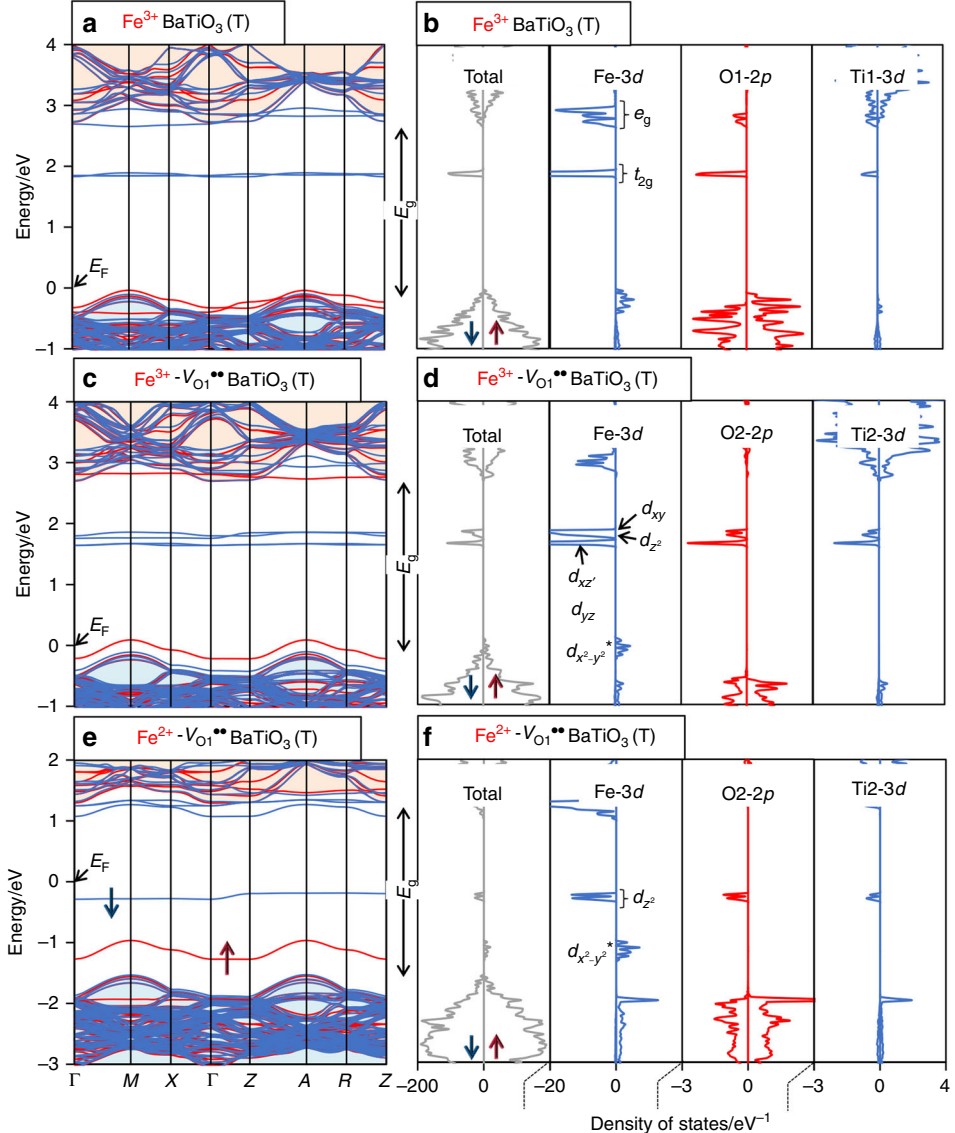

**Fig. 4 Electronic structures of tetragonal (T) Fe-BaTiO₃.** Band structures and total (partial) density of states (DOS) for the **a**, **b** $Fe^{3+}$, **c**, **d** $Fe^{3+}-V_{O1}^{\bullet\bullet}$, and **e**, **f** $Fe^{2+}-V_{O1}^{\bullet\bullet}$ cells, where $V_{O1}^{\bullet\bullet}$ denotes the oxygen vacancy on the O1 site. The Fermi-level is indicated by $E_F$. In the band structure, the red and blue lines represent the majority (↑) and minority (↓) spin components, respectively. The valence band and the conduction band of the host BaTiO₃ lattice are colored light blue and light orange, respectively. The bandgap is denoted by $E_g$. The red and blue arrows in **e** represent electrons occupying in their corresponding spin bands. In the DOS, the right and left panels correspond to the ↑ and ↓ bands, respectively, as indicated in the total DOS. The $Fe^{3+}$ cell has the empty gap state of $t_{2g}$ (bonding) in the ↓ band. In the $Fe^{3+}-V_O^{\bullet\bullet}$ and $Fe^{2+}-V_O^{\bullet\bullet}$ cells, the degeneracy of the $d$ orbitals is lifted owing to the presence of $V_O^{\bullet\bullet}$. In the $Fe^{3+}-V_O^{\bullet\bullet}$ cell, four empty $d$ (↓) states (bonding) appear near the middle of the bandgap. In the $Fe^{2+}-V_O^{\bullet\bullet}$ cell, the electron-occupied $d_{x^2-y^2}^*$ (↑) state is above the valence band maximum (VBM) (asterisk denotes antibonding), and the $E_F$ is formed near the middle of the bandgap by the occupied $d_{z^2}$ (↓) state (bonding).

above the second onset, $h^{\bullet}$ is generated in addition to $e'$. This yields $e'$-$h^{\bullet}$ pairs, thereby leading to an intense PV response. In the $h\nu$ range from the second to the third onset, the concentration of $h^{\bullet}$ is higher than that of $e'$. This is because the two orbitals of $Fe^{3+}$ and the three orbitals of $Fe^{3+}-V_O^{\bullet\bullet}$ act as an accepter state for $h^{\bullet}$ injection whereas the only one orbital of $Fe^{2+}-V_O^{\bullet\bullet}$ contributes to a donor state for $e'$ injection. We therefore consider that $e'$-$h^{\bullet}$ pairs and the surplus $h^{\bullet}$ contribute to the PV currents in this energy range. Above the third onset, $e'$ pumped from the $Fe^{2+}-3d_{x^2-y^2}^*$ (↑) state is superimposed; hence, the $e'$-$h^{\bullet}$ concentration increases markedly, thereby delivering the robust PV effect.

Figure 7a presents the current density ($J$) versus bias voltage ($V$) properties under light at $h\nu = 3.1$ eV. The oxidized sample exhibits

a short-circuit $J$ ($J_{sc}$) of $-32$ nA cm$^{-2}$ and an open-circuit voltage ($V_{oc}$) of 5.9 V. The reduced sample shows a large response: compared with the oxidized sample, the reduced sample has a $J_{sc}$ of $-520$ nA cm$^{-2}$ that is more than one order of magnitude and an extrapolated $V_{oc}$ of $\approx 35$ V that is approximately six times.

The current density $J_i$ arising from the bulk-PV effect is expressed by a third-rank tensor $\beta_{ijk}$. For the tetragonal crystal in 4 mm point group, the non-zero components of the bulk PV tensor are $\beta_{31}$, $\beta_{33}$, and $\beta_{15}$. Figure 7b indicates the data of $J_3$ ($\Theta$) at $h\nu = 3.1$ eV and the results of the fitting analysis, where $\Theta$ denotes the light-polarization angle. As listed in Supplementary Table 1, the oxidized sample has $\beta_{33} = -1.77 \times 10^{-9}$ V$^{-1}$ and $\beta_{31} = -2.86 \times 10^{-9}$ V$^{-1}$, while the reduced sample features larger

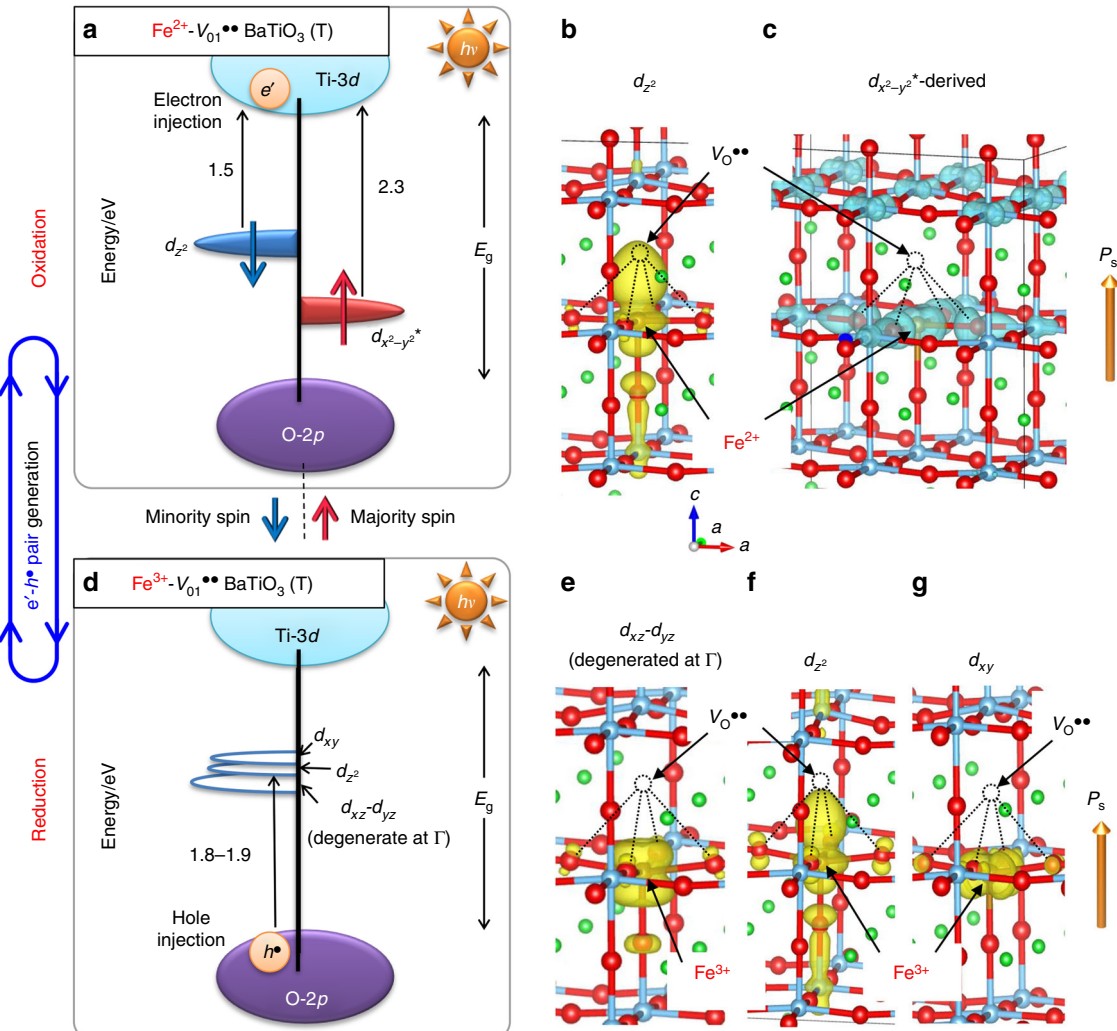

**Fig. 5 Mechanism of the visible-light PV effect.** The photoresponse of the reduced sample in the $Fe^{2+}$-$Fe^{3+}$ coexisting state is focused. Because of the charge neutrality and the attractive interaction, $Fe^{2+}$ is associated with an oxygen vacancy ($V_{O1}^{\bullet\bullet}$). Half of $Fe^{3+}$ also forms an $Fe^{3+}$-$V_{O1}^{\bullet\bullet}$ associate and the other half is isolated. The defect (gap) states are schematized for the **a** $Fe^{2+}$-$V_{O1}^{\bullet\bullet}$ and **b** $Fe^{3+}$-$V_{O1}^{\bullet\bullet}$ cells. The $k$-weighted and averaged energy differences are indicated by black arrows. The right and left states are the majority (↑) and minority (↓) spin components, respectively, and electrons occupying in the gap states are denoted by red and blue arrows in their corresponding spin bands. The partial charge densities near Fe atoms at the Γ point are displayed: the $Fe^{2+}$-$V_{O1}^{\bullet\bullet}$ cell has the **b** $d_{z^2}$- (↓) and **c** $d_{x^2-y^2}^*$ (↑) states, while the $Fe^{3+}$-$V_O^{\bullet\bullet}$ cell involves the **e** $d_{xz}$-$d_{yz}$ (↓) (degenerate), **f** $d_{z^2}$ (↓), and **g** $d_{xy}$ (↓) states. The partial charge densities in the ↑ and ↓ bands are colored light blue and yellow, respectively. The $Fe^{2+}$-$V_O^{\bullet\bullet}$ cell shows the electron-occupied $d_{x^2-y^2}^*$ (↑) state at a depth of 2.3 eV from the conduction band minimum (CBM) and the occupied $d_{z^2}$ (↓) state at 1.5 eV from the CBM. The $Fe^{3+}$-$V_O^{\bullet\bullet}$ cell contains the empty $d$ (↓) states at 1.8 to 1.9 eV from the valence band maximum (VBM). Light illumination induces electron ($e'$) injection from the $Fe^{2+}$-$d_{z^2}$ (↓) donor state along with an oxidation to $Fe^{3+}$ and hole ($h^\bullet$) injection from the empty $Fe^{3+}$-$d$ (↓) acceptor states that are associated with a reduction to $Fe^{2+}$. A sequential electron pumping from the valence band to the conduction band through the acceptor and donor states can, in principle, be realized at a photon energy ($h\nu$) much smaller than $E_g$. As $e'$-$h^\bullet$ pairs are created through two sequential redox reactions of $Fe^{3+}/Fe^{2+}$, the resultant PV response is termed the successive redox-mediated ferrophotovoltaics.

components by over an order of magnitude: $\beta_{33} = -3.25 \times 10^{-8}$ $V^{-1}$ and $\beta_{31} = -5.54 \times 10^{-8}\,V^{-1}$. We confirmed that $\beta_{15}$ is much smaller than $\beta_{31}$ and $\beta_{33}$ for both samples.

**Photo-luminescence analysis.** Figure 8 displays the photo-luminescence (PL) spectra of the single-crystal samples at 10 K, where the samples have a rhombohedral $R3m$ structure. We confirmed that the electronic structures are essentially the same for the tetragonal (Fig. 4) and rhombohedral (Supplementary Fig. 7) cells. The data of the oxidized sample (Fig. 8a) could be traced by a single log-normal function, and the long-wavelength edge (LWE) is

roughly estimated to be ≈660 nm (1.88 eV). This is in satisfactory agreement with the $Fe^{3+}$-derived PV onsets (1.9–2.0 eV; see Fig. 6a, b). The PL spectra of the reduced sample (Fig. 8b) could not be reproduced by a single function but are are well fitted by a superposition of two log-normal functions. The LWEs are evaluated to be ≈660 nm (1.88 eV) and ≈570 nm (2.18 eV), which agree quantitatively with the $Fe^{3+}$-derived second onset (1.9 eV) and the $Fe^{2+}$-derived third onset (2.3 eV), respectively (Fig. 6d). The cathode-luminescence (CL) analysis of Fe (1%)-doped $BaTiO_3$ ceramics (Supplementary Fig. 8) gave consistent results: LWEs of ≈680 nm (1.82 eV) for the oxidized ceramic and those of ≈630 nm

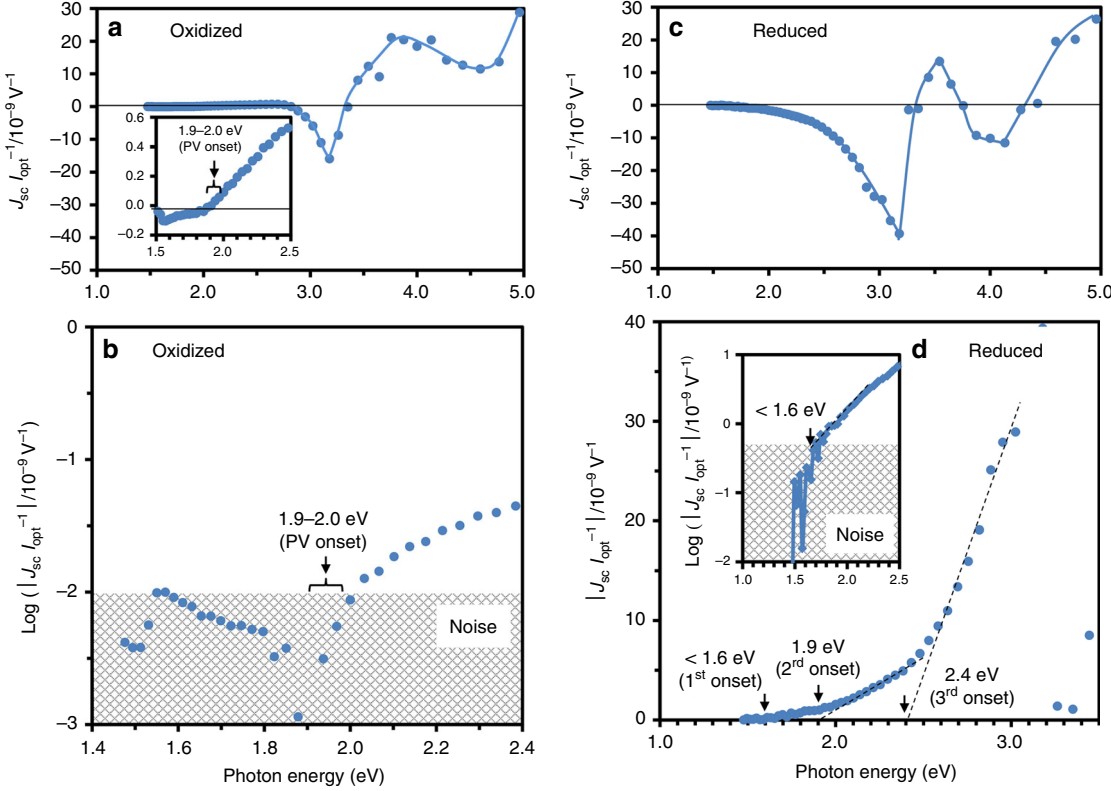

**Fig. 6 Photovoltaic currents versus photon energy.** The short-circuit current density ($J_{sc}$) normalized by optical intensity ($I_{opt}$) is plotted as a function of photon energy ($h\nu$) for the **a**, **b** oxidized and **c**, **d** reduced samples. The valence state of iron in the oxidized sample is $Fe^{3+}$ as the majority, while that in the reduced sample is a mixture of $Fe^{2+}$ and $Fe^{3+}$. The first PV onset is determined as the threshold by considering the noise level and the strong emissions of Xe lamp (light source) in the near-infrared range. The oxidized sample displays the first PV onset at 1.9–2.0 eV. The reduced sample exhibits a multiple-PV process: the first PV onset appears at less than 1.6 eV followed by the second at 1.9 eV and the third at 2.4 eV.

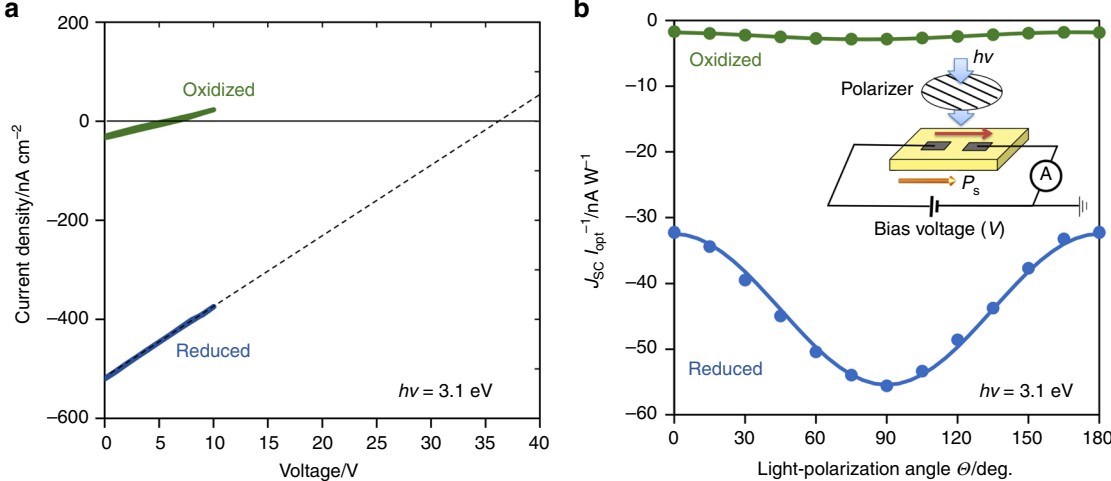

**Fig. 7 PV current density (J)-bias voltage (V) properties.** The measurements were performed under light at a wavelength of 405 nm (photon energy $h\nu$ of 3.1 eV) and an intensity of 5.6 W cm$^{-2}$. The J–V properties at a light-polarization angle ($\Theta$) of zero are presented in **a**, and the short-circuit current density ($J_{sc}$) versus $\Theta$ is plotted in **b** along with the measurement system. The inset in **b** schematizes the measurement system. The direction of $I_{opt}$ is set at $i=1$ and that of $P_s$ at $i=3$. The light-polarization ($\Theta$) is defined as the angle between the polarization plane of light and the measured direction of J. In the configuration in the inset of **b**, the photocurrent is parallel to $P_s$, and the current density is $J_3$. The fitting analysis of the $J_{sc}(\Theta)$ data by the equation derived from the symmetry consideration provides the bulk PV tensor elements of $\beta_{31}$ and $\beta_{31}$, as listed in Supplementary Table 1.

(1.97 eV) and ≈530 nm (2.34 eV) for the reduced ceramic. Moreover, the integrated data of the reduced ceramic indicate a CL peak with a LWE of ≈860 nm (1.44 eV), which agrees with the $Fe^{2+}$-derived first onset (<1.6 eV).

## Discussion

Our study demonstrates that the donor and acceptor states derived from iron ions activate the visible-light PV effect that is mediated through the successive $Fe^{3+}/Fe^{2+}$ redox cycles. Another

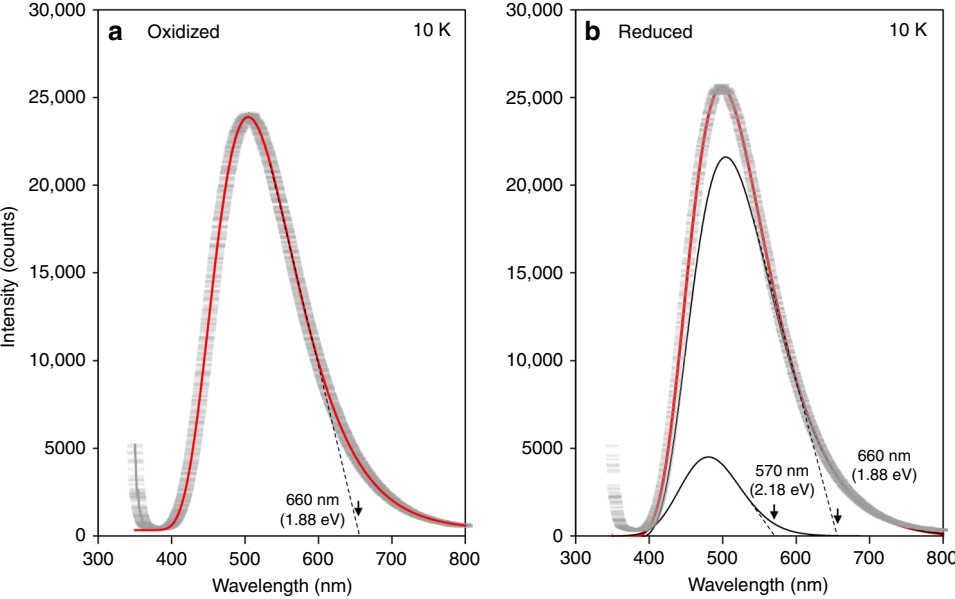

**Fig. 8 Photo-luminescence (PL) spectra.** The PL data (gray crossbar) were collected at 10 K for the single-crystal samples. The profile of the oxidized sample can be traced by a single log-normal function (red curve with $H = 23{,}548$, $\lambda_0 = 341.5$ nm, $w = 369.1$ nm, and $\rho = 1.50$). In contrast, that of the reduced sample could not be fitted by a single function and is well reproduced by a superposition (red curve) of two log-normal functions (black lines): the parameters of the large peak are $H = 21{,}720$, $\lambda_0 = 341.5$ nm, $w = 369.1$ nm, and $\rho = 1.50$, and those of the small peak are $H = 4{,}502$, $\lambda_0 = 148.0$ nm, $w = 263.4$ nm, and $\rho = 1.16$. The oxidized sample displays a long-wavelength edge (LWE) of $\approx$660 nm (1.88 eV), while the reduced sample exhibits LWEs of $\approx$660 nm (1.88 eV) and $\approx$570 nm (2.18 eV).

promising dopant is Mn that has the valence states of $2+$, $3+$, and $4+$ [57,58]. As shown in Supplementary Figs. 9 and 10, the mixed valence state of $Mn^{2+}$ and $Mn^{3+}$ can activate sequential redox reactions that yield $e'$-$h^\bullet$ pairs. The $Mn^{2+}$-$V_O^{\bullet\bullet}$ associate creates a donor state at a depth of 1.3 eV from the CBM, thereby giving rise to the first PV onset. Considering the isolated $Mn^{3+}$ having a acceptor state at 1.9 eV from the VBM, we expect that $e'$-$h^\bullet$ pairs can be generated at $\approx$1.9 eV, thereby leading to an intense PV response. The multiple gap states arising from Mn could produce the visible-light PV effect.

In summary, it is expected that our approach can be applied for photocatalysts of $TiO_2$ and $SrTiO_3$, as displayed in Supplementary Figs. 11 and 12, respectively. In these titanate systems, rigid $TiO_6$ octahedra govern the electronic structure near the bandgap, and the overall features of the iron-derived gap states are essentially the same. Moreover, the multivalent cations are not the unique dopants for $e'$-$h^\bullet$ pair generation. We can adopt a co-doping strategy: donor and acceptor states arise from cations of different elements, where their defect concentrations are easily controllable. Our work offers a starting point for further investigation of successive redox-mediated functions and opens an unprecedented route to robust photoinduced effects based on gap-state engineering.

## Methods

**Sample preparation and measurements.** An Fe (0.3%)-doped $BaTiO_3$ bulk single-crystal was grown by a top-seeded solution growth method. After cutting the crystals, we annealed them in air at 1250 °C for 12 h in air for recovery from mechanical damage. To control the valence state of iron, the reduced sample was annealed at an oxygen partial pressure at 900 °C ($Po_2^{900\,°C}$) of $10^{-23}$ atm for 100 h, which was followed by quenching to 150 °C, and was slowly cooled to room temperature. The oxidized sample was also prepared by annealing at 1200 °C at an $Po_2^{1200\,°C}$ of 0.2 atm in a similar manner. A commercial single-crystal of $BaTiO_3$ (NEOTRON CO., LTD.) was used as a reference. Platinum electrodes were fabricated on the sample surface by sputtering. As a poling treatment, we applied an electric field of 2 kV $cm^{-1}$ during a slow cooling from 150 °C to room temperature through the Curie temperature ($T_C \approx 130$ °C). This process enables us to obtain the

single-domain sample with $P_s//[001]$. Ceramics of Fe (1.0%)-doped $BaTiO_3$ were prepared by solid-state reaction for luminescence measurements.

**Photo- and cathode-luminescence measurements.** Photo-luminescence (PL) spectra of the single-crystal samples were measured at 10 K with a spectrometer (Horiba LabRam-HR PL) through a long-wavelength-pass filter (>325 nm). Data were collected with a He-Cd laser excitation at 325 nm (a spot size of $\approx$100 μm and a laser power of $\approx$1.7 mW). As the data at a wavelength ($\lambda$) above 800 nm were strongly influenced by an inherent optical property of the filter, we analyzed the PL intensity ($I_{PL}$) in the $\lambda$ range of 375–800 nm using the following log-normal function:

$$I_{PL} = H \cdot \exp\left[\frac{-\ln 2}{(\ln \rho)^2}\left\{\ln\left(\frac{(\lambda - \lambda_0)(\rho^2 - 1)}{w\rho}\right) + 1\right\}^2\right] \quad (1)$$

where $H$ denotes the peak height, $\lambda_0$ the shift parameter, $w$ the full-width at half maximum, and $\rho$ the half-width ratio. Cathode-luminescence (CL) spectra were also collected at 298 K with a scanning electron microscopy system (JEOL JSM-7800F Prime). Unfortunately, the CL intensities of the single-crystal samples were below the detection limit. Instead, we measured CL data of Fe (1.0%)-doped $BaTiO_3$ ceramics so that the output signal-to-noise ratio should be as high as possible. We analyzed the CL intensity ($I_{CL}$) data using the log-normal function in the similar manner.

We consider that luminescence profiles depend not only on the energy levels of defect states but also on the density-of-states of the relevant bands[59,60]. It is reasonable to estimate the energy levels of gap states with respect to the valence band maximum (VBM) or the conduction band minimum (CBM) from the long-wavelength edge (LWE) rather than the peak wavelength of the PL/CL luminescence profiles. However, the spectra exhibit an inherent tail on the long-wavelength side, which prevents us from determining the edge unambiguously. Instead, we evaluate the LWE from the linear slope (dashed lines in Fig. 8 and Supplementary Fig. 8) of the log-normal distribution.

**Photovoltaic analysis.** The current density $J_i$ is the component along the $i$ direction arising from the bulk-PV effect and is expressed by the following equation using a third-rank bulk PV tensor $\beta_{ijk}$:[11–13]

$$J_i = I_{opt}\beta_{ijk}e_je_k \quad (2)$$

where $I_{opt}$ denotes the light intensity and $e_j$ and $e_k$ are the components of the unit vectors along the $j$ and $k$ directions, respectively. We adopt the standard $3 \times 6$ matrix notation: $\beta_{ijk} \rightarrow \beta_{i\lambda}$ with $\lambda = 1, 2, \ldots, 6$, as used in the representation of the piezoelectric tensor. As the direction of $I_{opt}$ is $i = 1$, that of $P_s$ is $i = 3$, and the light-polarization ($\Theta$) is defined as the angle between the polarization plane of light and

the measurement direction of $J$ (Fig. 7b), the photocurrent densities in 4 mm point group[22] are expressed by

$$J_1 = 0 \left( //I_{opt} \right) \tag{3}$$

$$J_2 = I_{opt} \beta_{15} \sin 2\Theta (\perp P_s) \tag{4}$$

$$J_3 = I_{opt} \left[ (\beta_{33} + \beta_{31})/2 + (\beta_{33} - \beta_{31})/2 \cos 2\Theta \right] (//P_s) \tag{5}$$

We measured the current density ($J$)–bias voltage ($V$) characteristics under light with an intensity of 5.6 W cm$^{-2}$ using a laser module with a wavelength of 405 nm (a photon energy $h\nu$ of 3.1 eV). The light-polarization was controlled by a half-wavelength plate and a polarizer. The details are described in the preceding paper[22]. We investigated the normalized short-circuit current density $J_{sc} I_{opt}^{-1}$ as a function of $h\nu$. Monochromatic light from a Xe lamp passing through a MgF$_2$-coated diffraction grating was irradiated to the sample using an optical fiber. The wavelength resolution of our optical system was estimated to be 20 nm. Through the PV measurements, we excluded a transient current due to the capacitance and resistance of the samples together with a pyroelectric current caused by the photothermal[61] effect. In determining the open-circuit voltage ($V_{oc}$), we extrapolated the linear $J$–$V$ characteristics in a limited range owing to our current amplifier. All the PV measurements were performed at 25 °C.

**Defect concentration calculations**. The defect concentrations in an equilibrium state were calculated on the basis of the thermodynamic data set for BaTiO$_3$, the details of which are described in Supplementary Information. The calculations were conducted by a generalized-reduced-gradient non-linear least-squares method. We made the following assumptions: the lattice is equilibrated at 900 °C or 1200 °C (the annealing temperature) at a specific $Po_2$, and then the concentration of oxygen vacancy ($V_O^{\bullet\bullet}$) is quenched (fixed) at room temperature. We take into account the electronic reactions, such as the redox of Fe$^{3+}$/Fe$^{2+}$ and the thermally activated electron-hole pair generation of $null \leftrightarrow e' + h^\bullet$ in the whole temperature range.

**Density-functional theory (DFT) calculations**. DFT calculations were conducted using the generalized gradient approximation[62] with a plane-wave basis set. We used the projector-augmented wave method[63] as implemented in the Vienna ab initio simulation package (VASP)[64]. We employed the Perdew–Burke–Ernzerhof gradient-corrected exchange-correlation functional revised for solids (PBEsol)[65] and a plane-wave cut-off energy of 520 eV. Within the simplified generalized gradient approximation (GGA) + U approach[66], we added on-site Coulomb interaction parameters of $U-J$ of 2 eV to Fe-3$d$ and Mn-3$d$ throughout our calculations. We created TM-substituted cells (TM = Fe and Mn) with $3 \times 3 \times 3$ from the optimized BaTiO$_3$ cell with tetragonal $P4mm$ symmetry, leading to the tetragonal (T) Ba$_{27}$Ti$_{27}$O$_{81}$ structure. As displayed in Supplementary Fig. 2, there exists twenty-one types of oxygen atoms with different site symmetry. We therefore performed calculations for all the supercells with $V_{On}^{\bullet\bullet}$, where $V_{On}^{\bullet\bullet}$ denotes the oxygen vacancy on the $n$-th nearest-neighbor site with respect to Fe ions. Structural relaxation was performed for the atoms with less than 0.6 nm distance from the original position of $V_{On}^{\bullet\bullet}$. The valence states of Fe atoms were controlled by adjusting the total number of electrons. We employed the $\Gamma$-centered $3 \times 3 \times 3$ $k$-point mesh for the structural relaxations and the $5 \times 5 \times 5$ $k$-point mesh for density-of-states and band structure calculations. We added $U-J$ of 8 eV to Ti-3$d$ in addition to $U-J$ of 2 eV for Fe-3$d$ and Mn-3$d$ for the electronic structure calculations.

## Data availability.

The data that support the findings of this study are available upon request from the corresponding author.

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

## Acknowledgements
This research is supported by JSPS through Grant-in-Aid for JSPS Fellows (26-4693). This research is partly supported by JSPS KAKENHI Grant Numbers 26249094 and 17H06239. We thank Mr. Y. Shinozuka (MST CO., LTD.) for PL measurements and Mr. S. Ohtsuka for CL measurements.

## Author contributions
Y.N. conceived and initiated the project. M.M. directed the research. Y.N. carried out the theoretical study and wrote the manuscript. Y.T. and R.I. performed the experiments.

## Competing interests
The authors declare no competing interests.
