## [Peer Review File · Nature Communications]

Reviewers' comments:

Reviewer #1 (Remarks to the Author):

This manuscript reports a so-called successive-redox mediated ferrophotovoltaic which is shown to exhibit a stronger visible-light response than the parent material. The work includes both experimental and ab-initio calculations and suggests an important role for both Fe²⁺ and Fe³⁺ in BaTiO₃ to introduce donor and acceptor levels, respectively. In turn, it is proposed that two sequential Fe³⁺/Fe²⁺ redox reactions are active and enhance the photogenerated power not only under visible light but also at photon energies greater than the bandgap.

While addressing an interesting topic, this manuscript makes many assumptions and attributions without evidence to support them. For example, line 146-7, it's just stated that this oxidized Fe-doped BTO is the same as Mn-doped BTO from another paper. What evidence is given to prove this? This is just one of many such instance. All told, the manuscript feels underdeveloped – lacking in the details and careful measurements needed to assure the reader that the conclusions that are made are sound. Instead, it relies on previously held assumptions or statements that, in many cases, are not explicitly referenced, to make attributions without evidence to support it. In the current form then, the manuscript seems to lack the depth of evidence and support to assure the reader that the conclusions are sound and without greater attention to those points publication at this time might be considered in appropriate.

Additional comments:

- Lines 68-73 – References are needed to support the statements pertaining to the electronic configurations and positioning of the states. Similar points could be made throughout.
- Figures 1c and d are not useful in the current form. There too "schematic", probably would be better suited to a single figure to show the coupled nature of the effect proposed, and need to have references or data to support the placement of the inter-gap states.
- Figure 1e – Where is this data from? No details of how this data was produced are provided.
- Considerable assumptions about the defect structures are made in this paper. For example, in lines 80-95 alone 3-4 assumptions are made. How are the authors sure of the defect types – why doubly-charged oxygen vacancies vs. singly-charged vacancies? What evidence do you have of this and other defects types, concentrations, etc.? What about defect clusters?
- Figure 3a/d is a repeat (essentially) of Figure 1c and d. It seems redundant and unnecessary to have both in the manuscript.
- Figures 3b, c, e, f, and g are not used or called.
- For Figure 4, why isn't pristine BTO shown for a comparison in addition to the oxidized sample? Why isn't a reduced BTO shown for a comparison as well? Furthermore the attribution of 3 different onsets for the reduced samples is interesting – sure there is a clear one at 2.4 eV and at some point (1.6 eV) it turns on, but the effect at 1.9 eV is very slight. There are considerably better ways to measure the presence of inter-gap states (other optical methods CL/PL, DLTS, etc.) – it seems necessary to use these to confirm these things. At the same time, these effects are attributed to these specific energy levels and states related to the Fe doping, but there are many things happening when you anneal these materials in very low pressures of oxygen – why couldn't this be related to oxygen vacancies and clusters thereof, or clusters of vacancies and other species?
- Why is Figure 5 incomplete? Why is the data only taken to a voltage of ~10 V. Drawing a straight line to 35 V is again quite an assumption.

Reviewer #2 (Remarks to the Author):

The authors report on PV effects in Fe-doped BTO crystals. The bulk photovoltaic effect in this material has been investigated before and is well known, however, this work addresses an

important aspect and potential new concept regarding electronic states and associated photovoltaic charge separation in crystals with mixed valence of the iron dopant. The coexistence of donor and acceptor levels associated with the described two sequential Fe³⁺/Fe²⁺ redox reactions are quite interesting and could also apply to other materials. In my opinion this point warrants publication in Nature Communications.

A few technical comments:

- 1) Would it be possible to investigate the calculated energy states in the gap by some spectroscopy method, for example photoluminescence? This would make the presented claim much stronger.
- 2) English proof reading should be conducted.

Reviewer #3 (Remarks to the Author):

In the present manuscript, the authors investigate the photovoltaic properties of Fe-doped BaTiO₃ (BTO). They use DFT to calculate the electronic structure of crystals with isolated Fe³⁺ dopants, Fe³⁺-VO defect associates and Fe²⁺-VO defect associates, with special respect to band-gap states created by the doping. They find that the Fe³⁺ will act as an acceptor, releasing holes into the valence band under illumination at around 1.9 eV, whereas Fe²⁺ is a donor, giving electrons into the valence band from two different band-gap states at 1.5 eV and 2.3 eV below the conduction band minimum, respectively. Based on a mechanism of cyclic oxidation/reduction of the Fe²⁺/Fe³⁺ centers with illumination that results in combined light-induced hole and electron charge transport, the authors demonstrate that the photovoltaic properties can be drastically improved if the native concentration of Fe²⁺ and Fe³⁺ is of the same order of magnitude. This is achieved by reduction treatment at an oxygen partial pressure of 10⁻²⁰ atm at 900°C.

The topic of the manuscript matches a recently rekindled interest in anomalous photovoltaic effects and the use of ferroelectrics as wide-bandgap materials for solar energy conversion. It should therefore be interesting and important to many scientists working in the field of photovoltaics. With its combination of first-principle calculations and experiments, the presentation is strong and convincing; the results presented are novel. It has to be mentioned here that there are already reports in the literature on a change in the charge transport mechanism in differently doped BTO-based systems. It changes from hole-dominated in as-grown or oxidised BTO based on Fe³⁺ to electron-dominated in reduced BTO based on Fe²⁺ (e.g. A. Mazur et al., Appl. Phys. B 65, 481 (1997); U. van Stevendaal et al., Appl. Phys. B 63, 315 (1996)). However, these studies typically interpret the charge transport on the base of a one-center model, without taking into account the possibility to use Fe³⁺ for hole transport and Fe²⁺ for electron transport simultaneously. With the introduction of this new element, the manuscript represents an advance in understanding likely to influence thinking in the field. The older studies should still be pointed out in the present manuscript.

Apart from this, the manuscript can be recommended for publication in Nature Communications in its present form.

Response letter

To Reviewers

Thank you very much for your kind and useful comments on our paper. I appreciate all of your comments. To respond to each of the comments, we have made a major revision described below.

We have added the experimental data of the photo-luminescence (PL) and the cathode-luminescence (CL) to confirm the energy levels of the gap states in our samples: **Figure 8** (the PL spectra at 10 K) and **Supplementary Figure 8** (the CL spectra at 289 K). Also, we have added the following experimental and DFT results: **Figure 2** (the DFT results showing the formation of defect associates), **Figure 7** (the PV results; modified), **Supplementary Figure 1** (to express the formation of the bonding and antibonding Fe-3d derived states), **Supplementary Figure 2** (to show all types of oxygen sites in our DFT calculations), **Supplementary Figures 3 and 4** (to explain the electronic origin of the defect-associate formation), **Supplementary Figure 5** (to show and compare the PV results of the non-doped single-crystal sample), **Supplementary Figure 6** (the spectrum of the light source, the Xe ramp, in our measurement system), **Supplementary Figure 7** (the electronic structures of the rhombohedral BaTiO₃ lattice to compare the PL results at 10 K). The details of the defect-concentration calculations are described in Supplementary Information.

Also, to express the electronic states of the Fe-3d derived gap states explicitly, we have added information on the majority (\uparrow) and minority (\downarrow) spin components to all the states, e.g., as follows:

Gap states derived from Fe-3d

-----In the Fe³⁺ cell (Fig. 4a, b), the empty t_{2g} (\downarrow) state appears in the bandgap at a depth of 1.9 – 2.3 eV from the VBM. ----- The filled Fe-3d_{z²} (\downarrow) state (Fig. 5b) emerges in the middle of the bandgap, which is positioned at a depth of 1.5 – 1.9 eV from the CBM. --- As shown in Fig. 5c, the occupied Fe-3d_{x²-y²}* (\downarrow) state with significant dispersion is present. -----

Moreover, to distinguish between the bonding and antibonding states as a result of the hybridization between Fe-3d and its adjacent orbitals, we have put asterisk (*) to the antibonding states, e.g., as follows:

Strategy for visible-light activation

----- As displayed in Supplementary Fig. 1, the hybridization between Fe-3d and its adjacent orbitals provides the bonding states (t_{2g} and e_g) and the antibonding states (t_{2g}^* and e_g^*), as for the majority spin components. Hereafter, the Fe-3d derived states without an asterisk, such as t_{2g} and d_{xy} , are the bonding states, while those with asterisk (*), e.g., t_{2g}^* and d_{xy}^* , are the antibonding ones.

To Reviewer #1

Comment 1

While addressing an interesting topic, this manuscript makes many assumptions and attributions without evidence to support them. For example, line 146-7, it's just stated that this oxidized Fe-doped BTO is the same as Mn-doped BTO from another paper. What evidence is given to prove this? This is just one of many such instance. All told, the manuscript feels underdeveloped – lacking in the details and careful measurements needed to assure the reader that the conclusions that are made are sound. Instead, it relies on previously held assumptions or statements that, in many cases, are not explicitly referenced, to make attributions without evidence to support it. In the current form then, the manuscript seems to lack the depth of evidence and support to assure the reader that the conclusions are sound and without greater attention to those points publication at this time might be considered in appropriate.

Reply 1

We have revised our manuscript to remove assumptions as much as possible. As for photoinduced carriers, we have added the following sentence along with the important, pioneering papers (including the papers suggested by Reviewer #3) in **Strategy for visible-light activation**:

We therefore consider the following strategy for generating e^-h^+ pairs under visible light: Fe^{3+} plays the role of an electron acceptor that results in hole injection into the valence band^{38,39}, and Fe^{2+} acts as an electron donor that leads to electron injection into the conduction band⁴⁰⁻⁴².

In addition, we have added the experimental and calculation results associated with the detailed explanations including the photo-luminescence (PL) and cathode-luminescence (CL) (Figure 8 and Supplementary Figure 8), the formation of defect associates (Figure 2) along with Supplementary Figure 2 (all types of oxygen sites), and the electronic origin of the defect-associate formation (Supplementary Figures 3 and 4). Please see the revised (red-colored) parts in **Formation of defect associates ($\text{Fe}^{3+}-V_{\text{O}}^{\bullet\bullet}$ and $\text{Fe}^{2+}-V_{\text{O}}^{\bullet\bullet}$), Gap states derived from Fe-3d, and Visible-light PV properties, Photo-luminescence analysis.**

Comment 2

- Lines 68-73 – References are needed to support the statements pertaining to the electronic configurations and positioning of the states. Similar points could be made throughout.

Reply 2

As for the electronic configurations of iron ions, we have added the references as follows:

In O_h symmetry (Fig. 1a, b), the electronic configurations of iron in the high spin state are expressed as Fe^{3+} (d^5) with $t_{2g}^3(\text{up}) e_g^2(\text{up}) t_{2g}^0(\text{down}) e_g^0(\text{down})$ and Fe^{2+} (d^6) with $t_{2g}^3(\text{up}) e_g^2(\text{up}) t_{2g}^1(\text{down}) e_g^0(\text{down})$ ³⁶. In the BaTiO_3 lattice, the states of $t_{2g}^3(\text{up}) e_g^2(\text{up})$ are located near the bottom of the valence band³⁷, while it is probable that the states of $t_{2g}^0(\text{down})$ of Fe^{3+} (Fig. 1c) and $t_{2g}^1(\text{down})$ of Fe^{2+} (Fig. 1d) are present inside the bandgap by tuning their local structures.

We have added the detailed explanations regarding the majority and minority spin components along with the bonding and antibonding states for all the Fe-3d derived electronic states throughout in our paper.

Comment 3

- Figures 1c and d are not useful in the current form. There too “schematic”, probably would be better suited to a single figure to show the coupled nature of the effect proposed, and need to have references or data to support the placement of the inter-gap states.

Reply 3

We have modified Figure 1, where information on the redox $\text{Fe}^{3+}/\text{Fe}^{2+}$ reactions are omitted and the contents are reduced to explain our strategy; please see **Strategy for visible-light activation**.

Comment 4

- Figure 1e – Where is this data from? No details of how this data was produced are provided.
- Considerable assumptions about the defect structures are made in this paper. For example, in lines 80-95 alone 3-4 assumptions are made. How are the authors sure of the defect types – why doubly-charged oxygen vacancies vs. singly-charged vacancies? What evidence do you have of this and other defects types, concentrations, etc.? What about defect clusters?

Reply 4

The details of the defect-concentration calculations are described in Supplementary Information; please see **Defect Concentration Calculations**. As for the formation of the defect associates, we have added the DFT results in Figure 2 and Supplementary Figure 2, and Supplementary Figures 3 and 4, along with the detailed explanations in **Formation of defect associates ($\text{Fe}^{3+}-V_{\text{O}}^{\bullet\bullet}$ and $\text{Fe}^{2+}-V_{\text{O}}^{\bullet\bullet}$)**. As for singly-charged vacancies of oxygen, we have added the following explanations along with the important references in **Defect Concentration Calculations**:

The singly charged oxygen vacancy (V_{O}^{\bullet}) has been reported in the SrTiO_3 system, while the concentration of V_{O}^{\bullet} , $[V_{\text{O}}^{\bullet}]$, in Fe-doped SrTiO_3 is several orders of magnitude smaller than $[V_{\text{O}}^{\bullet\bullet}]^8$, where square brackets denote volumetric concentration, i.e., number of defects per cubic centimeter. As far as we are aware, the major contribution of V_{O}^{\bullet} has not been reported for BaTiO_3 ^{2,9}. We therefore consider only $V_{\text{O}}^{\bullet\bullet}$ as an oxygen vacancy throughout in this study. The singly charged oxygen vacancy (V_{O}^{\bullet}) has been reported in the SrTiO_3 system, while the concentration of V_{O}^{\bullet} , $[V_{\text{O}}^{\bullet}]$, in Fe-doped SrTiO_3 is several orders of magnitude smaller than $[V_{\text{O}}^{\bullet\bullet}]^8$, where square brackets denote volumetric concentration, i.e., number of defects per cubic centimeter. As far as we are aware, the major contribution of V_{O}^{\bullet} has not been reported for BaTiO_3 ^{2,9}. We therefore consider only $V_{\text{O}}^{\bullet\bullet}$ as an oxygen vacancy throughout in this study.

Comment 5

- Figure 3a/d is a repeat (essentially) of Figure 1c and d. It seems redundant and unnecessary to have both in the manuscript.

Reply 5

We have omitted the redox reactions of iron ions in Fig. 1. The description of the successive redox-mediated PV effect is shown only in Fig. 5 and we have revised the explanations; please see **Gap states derived from Fe-3d**.

Comment 6

- Figures 3b, c, e, f, and g are not used or called.

Reply 6

We have called all the figures in the manuscript including the partial charges in Fig. 5.

Comment 7

- For Figure 4, why isn't pristine BTO shown for a comparison in addition to the oxidized sample? Why isn't a reduced BTO shown for a comparison as well? Furthermore the attribution of 3 different onsets for the reduced samples is interesting – sure there is a clear one at 2.4 eV and at some point (1.6 eV) it turns on, but the effect at 1.9 eV is very slight. There are considerably better ways to measure the presence of inter-gap states (other optical methods CL/PL, DLTS, etc.) – it seems necessary to use these to confirm these things. At the same time, these effects are attributed to these specific energy levels and states related to the Fe doping, but there are many things happening when you anneal these materials in very low pressures of oxygen – why couldn't this be related to oxygen vacancies and clusters thereof, or clusters of vacancies and other species?

Reply 7

As mentioned above, we have added the experimental data of the photo-luminescence (PL) and cathode-luminescence (CL) to confirm the energy levels of the gap states in our samples. The DFT results along with the PV onset energies determined by the photocurrent measurements are in satisfactory agreement with the PL/CL analyses; please see **Visible-light PV properties** and **Photo-luminescence analysis**. In addition, the PV data of the non-doped single crystal sample have been added in Supplementary Fig. 5 and compared with the reduced and oxidized samples. We think that the impacts of oxygen vacancies and defect clusters (the defect associates) are addressed and explained in our revised manuscript. We do not address defect clusters including more than three point defects, because the purpose of this study is to propose novel materials design for visible-light activation of PV effects by using donor and acceptor states simultaneously.

Comment 8

- Why is Figure 5 incomplete? Why is the data only taken to a voltage of ~10 V. Drawing a straight line to 35 V is again quite an assumption.

Reply 8

Due to the limitation of the current amplifier in our system, we can measure PV data only at a limited voltage below 10 V. The explanations regarding this point have been added in **Method** and **Visible-light PV properties**, as follows:

Method

In determining the open-circuit voltage (V_{oc}), we extrapolated the linear $J - V$ characteristics in a limited range owing to our current amplifier.

Visible-light PV properties

The reduced sample shows a large response: compared with the oxidized sample, the reduced sample has a J_{sc} of -520 nA/cm^2 that is more than one order of magnitude and an extrapolated V_{oc} of $\sim 35 \text{ V}$ that is approximately six times.

To Reviewer #2

Comment 1

1) Would it be possible to investigate the calculated energy states in the gap by some spectroscopy method, for example photoluminescence? This would make the presented claim much stronger.

Reply 1

We have added the experimental data of the photo-luminescence (PL) and cathode-luminescence (CL) to confirm the energy levels of the gap states in our samples. The DFT results along with the PV onset energies determined by the photocurrent measurements are satisfactory agreement with the PL/CL analyses; please see **Visible-light PV properties** and **Photo-luminescence analysis**.

Comment 2

2) English proof reading should be conducted.

Reply 2

The manuscript has been revised according to the suggestions of English proofreading by American Journal Experts; please see the red colored parts.

To Reviewer #3

Comment 1

It has to me mentioned here that there are already reports in the literature on a change in the charge transport mechanism in differently doped BTO-based systems. It changes from hole-dominated in as-grown or oxidised BTO based on Fe³⁺ to electron-dominated in reduced BTO based on Fe²⁺ (e.g. A. Mazur et al., Appl. Phys. B 65, 481 (1997); U. van Stevendaal et al., Appl. Phys. B 63, 315 (1996)). However, these studies typically interpret the charge transport on the base of a one-center model, without taking into account the possibility to use Fe³⁺ for hole transport and Fe²⁺ for electron transport simultaneously. With the introduction of this new element, the manuscript represents an advance in understanding likely to influence thinking in the field. The older studies should still be pointed out in the present manuscript.

Reply 1

We have added the following sentence along with the important, pioneering papers including the papers suggested by Reviewer #3 in **Strategy for visible-light activation**:

We, therefore, consider the following strategy for generating e^-h^\bullet pairs under visible light: Fe³⁺ plays the role of an electron acceptor that results in hole injection into the valence band^{38,39}, and Fe²⁺ acts as an electron donor that leads to electron injection into the conduction band⁴⁰⁻⁴².

REVIEWERS' COMMENTS:

Reviewer #1 (Remarks to the Author):

I have read the rebuttal and revised manuscript. The vast majority of my concerns have been addressed and the manuscript is improved.

Reviewer #2 (Remarks to the Author):

Previous comments have been addressed satisfactorily.

Reviewer #3 (Remarks to the Author):

The authors have fully addressed the point raised in my previous report. Therefore, I can recommend to publish this manuscript in its present form. I have also read the comments made by the other reviewers and the reply by the author, but this does not make me want to change my assessment.

Response letter

To Reviewers

Thank you very much for your comments on our paper.

To Reviewer #1

Comment 1

I have read the rebuttal and revised manuscript. The vast majority of my concerns have been addressed and the manuscript is improved.

Reply 1

I appreciate the reviewer's comment.

To Reviewer #2

Comment 1

Previous comments have been addressed satisfactorily.

Reply 1

I appreciate the reviewer's comment.

To Reviewer #3

Comment 1

The authors have fully addressed the point raised in my previous report. Therefore, I can recommend to publish this manuscript in its present form. I have also read the comments made by the other reviewers and the reply by the author, but this does not make me want to change my assessment.

Reply 1

I appreciate the reviewer's comment. We are encouraged by this comment of Reviewer #3.